# Examining Body Appreciation and Disordered Eating In Adolescents of Different Sports Practice: Cross-Sectional Study

**DOI:** 10.3390/ijerph17114044

**Published:** 2020-06-05

**Authors:** Rasa Jankauskiene, Migle Baceviciene, Laima Trinkuniene

**Affiliations:** 1Institute of Sport Science and Innovations, Lithuanian Sports University, 4221 Kaunas, Lithuania; rasa.jankauskiene@lsu.lt; 2Department of Physical and Social Education, Lithuanian Sports University, 44221 Kaunas, Lithuania; laima.trinkuniene@lsu.lt

**Keywords:** body image, eating disorders, adolescent, exercise, health behaviour

## Abstract

This cross-sectional study aimed to examine the associations between body appreciation, body functionality and disordered eating in a large adolescent sample of different levels and types of sports practice. Method: The sample consisted of 1412 adolescents (59.8% were girls). The ages ranged from 15 to 18 years old, with a mean age of 16.9 (SD = 0.5) for girls and 17.0 (SD = 0.4) for boys. Participants completed an anonymous questionnaire assessing the nature of sports participation, body appreciation, body functionality, self-esteem, body dissatisfaction, internalisation of sociocultural beauty ideals and disordered eating. A two-way ANOVA was employed to test the differences in body image concerns, body appreciation, perceived physical fitness and disordered eating behaviours in gender groups and groups of different sport types and levels. Multiple linear regression analyses were performed to predict disordered eating behaviours of different study variables. Results: Participants of leisure and competitive sports reported greater body appreciation, self-esteem and lower body dissatisfaction compared to non-participants. No differences in body appreciation and disordered eating were observed in adolescents involved in weight-sensitive and less weight-sensitive sports. Body appreciation and body functionality were associated with lower disordered eating in adolescent girls not participating in sports, leisure exercisers and participants of competitive sport as well as in boys participating in competitive sports, controlling for body mass index. Adolescent boys demonstrated greater body appreciation compared to girls. Conclusions: The results of this study support the knowledge of the protective role of positive body image preventing dysfunctional eating in adolescent girls of various sports practice and in competitive sports involved boys. Disordered eating prevention and clinical treatment programs for adolescents of different physical activity might benefit from including education about body appreciation and functionality.

## 1. Introduction

The preventions of overweight and obesity and disordered eating and eating disorders in adolescents are issues of great importance. Adolescent disordered eating and obesity traditionally have been linked to negative body image [1,2,3]. Studies of adolescent samples have demonstrated that the negative facets of body image, such as body dissatisfaction, drive for thinness or muscularity, overweight preoccupation, body shame and self-objectification are associated with poorer psychological and physical health, disordered eating and overweight development in future life [4,5,6,7,8,9,10]. Intensive efforts have been produced to create various body image concerns and disordered eating prevention programs for adolescents [11,12]. Traditionally, these programs have been aimed to decrease body image concerns and body dissatisfaction. However, researchers have argued that focusing on decreasing the symptoms of negative body image without considering how to promote positive body image has limited scientific understanding of body image and prevents the development of effective intervention programs [13,14].

Positive body image consists of several main constructs, such as body appreciation, body pride and body acceptance [13]. The central facet of positive body image is body appreciation, which is defined as accepting, holding favourable opinions toward and respecting one’s body, resisting the sociocultural pressures to internalise the stereotyped beauty standards as the only form of human beauty and appreciating the functionality and health of the body [15,16]. Studies of college-aged women have demonstrated that positive body image is associated with greater self-esteem and proactive coping [17]. Men typically report greater body appreciation compared to women [18,19]. 

However, the development of positive body image is highly understudied in adolescent samples. This is somewhat surprising considering that adolescence is a critical period in the development of positive and negative body image. Nevertheless, some rare studies on adolescents have replicated findings in adults, concluding that adolescent body appreciation is inversely associated with body dissatisfaction, drive for thinness, sociocultural beauty ideals internalisation and social physique anxiety, and is positively associated with life satisfaction, self-esteem, physical well-being and intuitive eating [20,21,22,23]. Understanding the protective nature of positive body image is important to more deeply investigate the associations between positive body image and a health-related lifestyle, especially participation in physical activity and sports [12,24]. 

The greater participation in physical activity is associated with more favorable body image [24,25]. Nevertheless, the associations between adolescents’ positive body image (body appreciation) and physical activity are highly understudied [23]. The majority of studies explaining the associations between body appreciation and physical activity were implemented in adults participating in yoga [26,27,28]. Therefore, it is important to test these associations in adolescents participating in various leisure and competitive sports activities. The results of the present study might be important for physical activity promotion in adolescence. 

Additionally, a lack of research exists specifically in regard to the potential relationships between positive body image and physical activity-related variables such as type of exercising (organised leisure exercising versus competitive athletics). Studies on adolescents have demonstrated that sports-involved adolescents report greater body image compared to non-exercisers [29,30]. Nevertheless, involvement in leisure fitness activities that emphasise appearance, such as dance, gym sports and other fitness clubs’ organised activities, might compromise adolescent girls’ body images [30]. In contrast, based on the developmental theory of embodiment [31], some recent studies have demonstrated that yoga can provide young women with the opportunity to cultivate a more favourable relationship with their bodies [27,32]. Therefore, it is important to understand the associations between the positive body image and physical activity in adolescents practicing sports of different natures and aims [23,24]. Moreover, little is known of what role positive body image plays in the prevention of disordered eating in sports practice-involved adolescents, especially boys [26]. 

Studies in athletes concluded that athletes report more favourable body image compared to non-athletes [33]. Despite the recent attention to positive body image, to the best of our knowledge, only a few studies have assessed aspects of positive body image among athletes [34,35]. A recent study demonstrated that adult student athletes report greater levels of body appreciation compared to non-athletes and that positive body image is associated with greater sports confidence and flow-state in physical activity and more successful sport performance [34]. Thus, this study was aimed to provide more knowledge on positive body image in adolescent athletes of different sports experience. 

Prevention of disordered eating and eating disorders is one of the most important issues for competitive adolescent athletes, especially those competing in weight-sensitive sports in which body weight has a high impact on performance [36]. It is commonly agreed that practicing aesthetic, weight class, gravitational technical (i.e., high jump) and gravitational endurance (i.e., swimming) sports is related to greater risk of developing disordered eating [37,38]. Disordered eating typically starts at age 14–17-years-old, and this period overlaps with the time when athletes begin specialising in a particular type of sport [38]. Deep specialisation in a weight-sensitive sport might create distress for the adolescents undergoing major bodily changes, especially girls [38]. Since positive body image is associated with more favourable eating and weight control-related attitudes and behaviours [13], understanding the role of positive body image in the prevention of disordered eating of adolescent competitive athletes is an important scientific issue. The findings of this study might have important practical implications for the prevention of the dysfunctional eating behaviours in sports-involved adolescents. 

The main aim of this study was to explore the associations between the positive body image and disordered eating in a large adolescent sample of different levels and types of sports practice. Based on the previous findings we hypothesised that sports-involved adolescents will demonstrate greater body appreciation, body functionality, self-esteem and lower body dissatisfaction and sociocultural attitudes towards appearance, controlling for body mass index (BMI). Furthermore, we expected that adolescents involved in weight-sensitive sports would demonstrate greater disordered eating and lower body appreciation. Next, we assumed that body appreciation would be greater in adolescent boys compared to girls, controlling for BMI. Finally, we expected that body appreciation and body functionality would be associated with lower disordered eating behaviours in adolescents of different sports practice, controlling for gender and BMI.

## 2. Methods

The sample consisted of 1412 adolescents (59.8% were girls). The cluster random sample was formed by inviting to participate secondary schools from 26 Lithuanian cities. There were 41 randomly selected schools that participated in the study. In the selected schools, all students of the 11th year were invited to participate.

The mean age of the study sample was 16.9 (SD = 0.5) years for girls and 17.0 (SD = 0.4) for boys, and the age range was from 15 to 18 years-old. Of the study participants, 92.4% were 17 years old. The self-reported BMI of the sample ranged from 14.0 to 41.7 kg/m^2^, and the mean BMI for girls was 21.0 (SD = 3.0) kg/m^2^ and for boys, 22.0 (SD = 3.1) kg/m^2^. 

### 2.1. Procedure

This study was a part of a larger study based on an interdisciplinary approach and aimed to investigate associations between body image concerns and lifestyles in Lithuanian adolescents. Consent of school directors and parents was obtained, which gave permission for students to participate in this study. Respondents provided their answers by filling the questionnaires consisting of a battery of self-report questionnaires designed to measure study variables. There were 1492 students who were offered to participate in the study, 56 from them refused to participate in the study. Information on refusal reasons was not collected. Furthermore, 24 questionnaires were excluded from the analysis since they were incomplete. For the final analysis, 1412 questionnaires were confirmed.

### 2.2. Measures

The Body Appreciation Scale-2 (BAS-2) [15] was used to assess three facets of the body appreciation (body acceptance, respect for one’s body and resistance to pressure from the media’s appearance ideals). The instrument consists of 10 items rated on a five-point Likert-type scale (1 = Never, 5 = Always). Higher scores indicated greater body appreciation. The Lithuanian version of the scale demonstrated good psychometric properties [39]. The internal consistency of the scale in this study was 0.97.

The Rosenberg Self-Esteem Scale (RSES) [40] was used to assess global self-esteem and general feelings of self-worth. The scale is comprised of 10 items scored on a four-point Likert scale ranging from 1 (strongly disagree) to 4 (strongly agree). A higher score indicates a greater level of self-esteem. Cronbach’s alpha for the RSES in this study was 0.86.

The Body Dissatisfaction (BD) subscale from the Eating Disorder Inventory-3 [41] was used to assess body dissatisfaction. The body dissatisfaction scale consists of 10 items with Likert-type answers from always (4) to never (0). The subscale assesses dissatisfaction with overall body shape and size and particular parts of the body. Greater scores indicate greater body dissatisfaction. The subscale has adequate psychometric qualities in adolescent and young adult non-clinical samples [42,43]. In our sample, the internal consistency of the body dissatisfaction subscale was Cronbach α = 0.82.

The Self-Objectification Questionnaire (SOQ) [44] was used to assess the manner in which a person sees his/her body in an objectified, appearance-related manner or in a non-objectified, body functionality-related manner. Adolescents rank 10 body-related attributes in order of importance to them. Of the items, five are associated with appearance-based attributes (physical attractiveness and measurements, sexuality, firm/sculpted muscles and body weight), and the other five attributes were related to body functionality-based physical attributes (physical fitness, physical coordination, health, strength and energy level). Furthermore, all attributes are ranked starting from 1 to 10, where 1 is least important and 10 is most important). The appearance-related attribute rankings were summed together for one total, and the body functionality-based ones formed the second sum. Next, the sum for body functionality-based attributes was subtracted from the sum for the appearance-based ones. The final score ranged between −25 to 25, with the higher score indicating greater body objectification. In this study, we used the body functionality subscale only.

The Eating Disorder Examination Questionnaire 6.0 (EDE-Q 6.0) [45] was used to assess disordered eating. The EDE-Q provides a comprehensive evaluation of the essential behavioural and attitudinal characteristics of eating disorders and disordered eating behaviour. It is comprised of 28 items. First, the six open-ended questions resulted in frequency data on the essential behavioural characteristics of disordered eating: objective binge eating, self-induced vomiting, laxative use and excessive exercise. Furthermore, 22 attitudinal questions comprising four subscales reflect the severity of disordered eating characteristics. The answers are ranged on a six-point Likert scale from 0 (no day) to 6 (every day). A higher score indicates either greater severity or frequency. In this study, we used the general score of the questionnaire. The Lithuanian version of the scale demonstrated good psychometric properties [46]. Internal consistency for the general scale was good (α = 0.95). 

The Sociocultural Attitudes Towards Appearance Questionnaire-4 (SATAQ-4) [47] was used to assess the general role of sociocultural influences on body image and appearance-related internalisation. The original SATAQ-4 comprised five subscales (pressures from peers, family and media, internalisation thin/low body fat subscale, and the internalisation muscular/athletic subscale), and each of them is composed of items that are rated on a five-point Likert scale, where 1 means definite disagreement and 5 means definite agreement. The higher the score, the greater the acceptance or internalisation of the dominant sociocultural standards for appearance. The Lithuanian version of the questionnaire demonstrated excellent psychometric characteristics [48]. In this study, we used the total SATAQ-4 score. Internal consistency for the total score was good (α = 0.92). 

The Drive for Muscularity Scale (DMS) [49] was used to assess the attitudes and behaviours that reflect the preoccupation with an athletic body type and muscularity. The scale is aimed at exploring an individual’s perceptions of not being muscular enough and needing to increase their muscle mass and bulk their body frame irrespective of their actual muscle mass. The scale consists of from 15 items rated on a six-point Likert-type scale ranging from 1 (never) to 6 (always). Greater scores indicate stronger muscle development-related attitudes and more intensive behaviours. The Lithuanian version of the scale demonstrated good psychometric properties [50]. In this study, the internal consistency of the scale was good (α = 0.95). 

Body Mass Index (BMI) was assessed using self-reported heights and body weights. As recommended by the International Obesity Task Force (IOTF) cut-offs, the sample was classified into four body mass categories according to percentiles: below the 5th percentile was thin, between the 5th and 84th was normal weight, between the 85th and 94th was overweight and ≥the 95th percentile was obese [51]. 

The Leisure-time Physical Activity Questionnaire (LTEQ) [52] was used to assess physical activity of adolescents. This instrument measures weekly physical activity in three intensities: mild, moderate and strenuous. The number of bouts of mild exercise is multiplied by 3, moderate exercise by 5, and strenuous exercise by 9, resulting in a final score of physical activity. The final score of physical activity shows a total metabolic equivalent by each intensity level. A greater score indicates greater physical activity of a particular level.

Participation in a competitive sport was assessed using a single question: ‘Do you participate actively in competitive sports and take part in sports competitions with professional sports goals’ (answer: ‘yes/no’). Participation in leisure-time sports was assessed by a single question: ‘Do you exercise during leisure-time sports activities without the intention to compete in sports competitions?’ (answer: ‘yes/no’). For those who answered positively, a list of different kinds of sports was presented with the request to select one for competitive sports and two at the maximum for leisure-time sports. Each of the competitive and leisure-time sports was divided into two groups (weight-sensitive and less weight-sensitive sports) according to the proposed classification [53,54]. As leisure-time exercisers were provided a possibility to indicate two kinds of sports, the additional category of mixed sports was created for those who participated in both weight-sensitive and less weight-sensitive sports. The distribution of the competitive and leisure-time sports in weight-sensitive and less-weight-sensitive sports is presented in Appendix A.

Perceived physical fitness (PPF) was assessed by a single self-developed question “How do you evaluate your own fitness level compared to your classmates and/or friends?” Possible answers were: ‘very fit’, ‘fit enough’, ‘average fitness’, ‘a little unfit’ and ‘very unfit’. This question was used in previous studies [55].

### 2.3. Statistical Analysis

First, descriptive statistics of the sample were performed, and the normality of continuous variables was tested. For normally distributed variables, the comparison between two groups was performed using an independent samples t-test, and for non-normally distributed data, a Mann–Whitney U test was used. A one-way ANOVA with Bonferroni correction was used for multiple comparisons across groups of different levels and types of sport practice. Association between categorical variables was tested by the chi-square test. Cronbach’s alpha coefficients were used for the evaluation of internal consistency. A score of ≥0.90 was considered excellent. A two-way ANOVA was employed to test the effect of gender and different sports practice on body image concerns, body appreciation, perceived physical fitness and disordered eating behaviours, with body mass index as a covariate. Partial eta squared was calculated to represent the estimate of the effect size. Multiple linear regression analysis was performed to predict disordered eating behaviours separately in groups of different levels and types of sports practice separately in subsamples of boys and girls. Body appreciation, self-esteem, body dissatisfaction, drive for muscularity, sociocultural pressures towards appearance, body functionality and body mass index were entered as independent variables. The statistical analyses were carried out using IBM SPSS Statistics 26 (IBM Corp., Armonk, NY, USA). 

### 2.4. Ethical and Legal Aspects

Prior to the study’s beginning ethical approval by the Committee for Social Sciences Research Ethics of the Lithuanian Sports University was obtained (protocol No. SMTEK-32, 27-09-2019). No any personal information was collected, thus anonymity was ensured. The students were provided a possibility to select the option “I agree to participate” or “I disagree to participate” to give their consent to participate in the study before beginning the survey. Following the Helsinki Declaration ethical and legal principles of the research, the students were introduced to the aim of the study before the questionnaires were provided. The laws of anonymity and goodwill were followed. Information guidelines for the observational study in epidemiology have been followed to carry out the study and write the manuscript [56]. 

## 3. Results

### 3.1. Descriptive Analysis

Sample characteristics in different types and levels of sports practice are presented in Table 1. Girls composed a higher proportion of non-participants in sports, whereas boys made up a higher proportion of the participants in competitive sports. Normal weight was more prevalent in the competitive sports group, while underweight and overweight were more common in the non-participants in sports group. The scores of perceived physical fitness status and leisure-time physical activity were higher in sports participants. Multiple comparisons confirmed significant differences of physical fitness and physical activity scores across all three groups of different sports practices. Time of weekly exercise and the duration of involvement in sports in years was longer in the competitive sports group compared to the leisure sports group.

### 3.2. Body Image, Disordered Eating Behaviours and Lifestyle-Related Factors in Sports Groups

Table 2 and Table 3 represent two-way effects of gender and different sports practice on body appreciation, self-esteem, body image concerns, disordered eating behaviours and perceived physical fitness in boys and girls. In girls, body image concerns and disordered eating behaviours were higher, whereas body appreciation and self-perceived physical fitness were lower compared to boys. Gender did not affect the self-esteem and body functionality scores. Interesting findings were revealed when testing the effect of different sports practices. Body appreciation, self-esteem, perceived physical fitness and sociocultural attitudes towards appearance were higher in adolescents participating in sports, especially in competitive sports. Boys not involved in any sports demonstrated higher body dissatisfaction scores. Gender and participation in different sports practice interaction effects were not observed.

### 3.3. Body Image, Disordered Eating Behaviours and Lifestyle-Related Factors in Weight Sensitive and Less Weight Sensitive Sports

Analysis of gender and participation in weight-sensitive vs. less weight-sensitive leisure-time sports interaction effects on body image concerns revealed significant differences for body dissatisfaction but not for body appreciation (Table 4). Girls participating in different weight-sensitive sports demonstrated similar body dissatisfaction scores. In boys, body dissatisfaction scores were higher in less weight-sensitive sports as compared to weight-sensitive. Interaction effect between gender and weight sensitivity was observed (F = 3.4, η^2^ = 0.009, *p* = 0.033). In the competitive sports group, statistically significant differences in body dissatisfaction scores between weight-sensitive and less weight-sensitive sports groups were observed. Girls and boys demonstrated higher body dissatisfaction in the less weight-sensitive sports group. 

Testing the hypothesis about disordered eating and sociocultural attitudes towards appearance differences in weight-sensitive and less weight-sensitive sports groups it was found that scores were higher in girls compared to boys, but no effects of weight sensitivity were observed. Also, the effect of gender was observed in drive for muscularity scores with no significant difference comparing weight-sensitive and less sensitive sports groups.

Perceived physical fitness scores were higher in boys as compared to girls. Besides, physical fitness was assessed better in weight-sensitive sports group in leisure exercisers as well as in adolescents involved in competitive sports as compared to less-weight sensitive sports.

### 3.4. The Associations between Study Variables and Disordered Eating in Sports Groups

To test the hypothesis that body appreciation has a preventive effect on disordered eating, a series of multiple regression analyses were conducted, controlling for body mass index. As can be seen in Table 5 and Table 6, a higher score of body appreciation and body functionality decreased the risk of disordered eating in all girls and boys participating in competitive sports. In contrast, perceived sociocultural pressures and body dissatisfaction were associated with increased risk of dysfunctional eating independent of sports practice of either gender. Moreover, a higher level of self-esteem demonstrated a protective effect on disordered eating in girls not involved in any sports.

## 4. Discussion

The main aim of this study was to explore the associations between positive body image and disordered eating in a large sample of adolescents of different sport practice. Based on the previous findings, we hypothesised that different sports activities-involved adolescents will demonstrate greater body appreciation, body functionality and self-esteem and lower body dissatisfaction, controlling for BMI and gender. No assumptions for the drive for muscularity, sociocultural attitudes towards appearance and disordered eating were developed. Our first assumption was partially confirmed. Leisure and competitive sports-involved adolescents reported significantly greater body appreciation compared to adolescents not involved in sports. This finding is in line with a study on young adults that demonstrated greater body appreciation in athletes compared to non-athletes [34]. This study adds important knowledge that involvement in leisure exercise is also important for greater positive body image in girls and boys. The differences in body appreciation between groups might be explained by increased body satisfaction through elevated levels of physical fitness [57] that was observed between groups in this study. Furthermore, the findings may also suggest that the associations between sport participation and positive body image might be mediated by the processes of increased embodiment and decreased self-objectification [58].

The present study demonstrated that sports-involved adolescents report significantly lower body dissatisfaction compared to adolescents not involved in sports. This finding overlaps with other studies that demonstrate that individuals participating in sports and physical activity report lower scores on a range of measures capturing negative body image [24]. Adolescent sport participants might experience lower distress associated with body image as they may objectively come closer to the sociocultural ideals of appearance than their non-exercising counterparts [38]. However, in the present study showed that competitive and leisure sports involved-adolescents report greater internalisation of sociocultural ideals towards appearance compared to non-exercisers. These results might be explained by the Tripartite Influence model of Thompson (1999) that states that internalisation of beauty ideals is an outcome of pressures from parents, peers and the media [59]. Sports involved-adolescents might experience pressures from their coaches as well; therefore, they may be faced with even greater pressure to attain beauty ideals and to maintain a particular body weight and/or shape than adolescents not involved in sports [60].

As expected, in this study, we found that leisure and competitive sports involved-adolescents report significantly greater self-esteem. A plethora of studies have demonstrated positive associations between the physical activity and self-esteem using an exercise and self-esteem model [61] that has recently been used to explain the associations between physical activity and self-esteem using mediators such as perceived physical fitness and more favourable body image [62,63].

Next, this study demonstrated that sports-involved adolescents report significantly greater drive for muscularity. Despite the findings that the drive for muscularity is associated with greater disordered eating [64], it should be noted that not all behaviours assessed by the drive for muscularity scale are damaging to health. For example, exercising to gain weight and eating products with high protein might also be assessed as healthy behaviour. Therefore, sports-involved adolescents might more frequently report this behaviour as part of their daily sports routine.

We did not find significant disordered eating differences between sports-involved adolescents and adolescents not participating in sports. These results coincide with other findings that demonstrate no disordered eating differences between adolescents of different sports practice [65,66,67] yet contradict studies that demonstrate the opposite [53,68]. However, in this study, we did not assess the status of competitive participation (elite athletes versus non-elite athletes), which might influence the results and be considered as a limitation. Future studies should address this issue. Thus, the findings on the first assumption demonstrated that competitive and leisure sports-involved adolescents demonstrate more favourable body image (greater body appreciation and lower body dissatisfaction) and self-esteem compared to their non-exercising counterparts despite the persistent and relatively greater sociocultural beauty ideals internalisation and the drive for muscularity. Future studies are recommended to more deeply understand the mechanisms underlying these associations.

Next, in the present study we hypothesised that adolescents involved in weight-sensitive sports would report greater disordered eating and lower body appreciation. This hypothesis was fully rejected. This study demonstrated that there are no differences in disordered eating between weight-sensitive and less weight-sensitive groups of adolescents participating in leisure-time and competitive sports. These findings are in line with other studies of adolescents [69]. This might be explained by the facts that athletes tend to under-report disordered eating and adolescent athletes face a shorter period of exposure to weight-sensitive sport-specific demands compared to adult athletes [37]. However, this is the cross-sectional study, and the causality of the associations is unclear. There were no differences in body appreciation observed. It seems that participation in weight-sensitive sports is associated with lower body dissatisfaction but not greater body appreciation. Nevertheless, as it is one of the first studies assessing adolescent body appreciation and participation in sports according to weight sensitivity, generalisation of findings is limited, and future studies should test these findings.

Furthermore, we assumed that body appreciation would be greater in adolescent boys compared to girls, controlling for BMI and different sports practice. This assumption was confirmed. Overlapping findings from the previous studies, this study demonstrated that boys report greater body appreciation compared to girls [20,21,22]. In line with previous studies, we found gender differences of negative body image in the expected direction, with females demonstrating greater body image concerns compared to boys [70].

Finally, we expected that body appreciation and body functionality would be associated with lower disordered eating behaviours in adolescents of different sports practice, controlling for gender, BMI and body dissatisfaction. This hypothesis was partially confirmed. In general, body appreciation was significantly associated with lower disordered eating in non-exercising, leisure exercising girls and competitive sports-involved adolescent girls and boys, controlling for BMI and body dissatisfaction. Body functionality was associated with lower disordered eating in non-exercising and leisure exercising girls, as well as in competitive sports involved boys. This evidence overlaps with previous findings in adolescents that demonstrate negative associations between body appreciation and disordered eating [20] and supports the assumption that positive body image is likely to be protective of healthier eating behaviours [13,22].

Our study contributes the important novel finding that body appreciation is associated with lower dysfunctional eating in girls of different sports practice and boys involved in competitive sports, controlling for BMI and body dissatisfaction. Despite the fact that body dissatisfaction and internalisation of the sociocultural attitudes towards appearance were significantly associated with disordered eating in girls of different sports practice, body appreciation remained a significant predictor of lower disordered eating. These findings have important implications that prove the importance of the development of positive body image in adolescent girls of different physical activity. This study adds to the knowledge that it is not enough to decrease body dissatisfaction and/or internalisation of sociocultural beauty ideals when preventing disordered eating. It is important to develop positive body image in intervention programs tackling disordered eating in general and sports involved-adolescents.

The present study expands the knowledge on the associations between positive body image and disordered eating in adolescents of different sports practice. It seems that body appreciation and functionality are important facets of positive body image that might help to prevent disordered eating in adolescent girls of different sports practice and boys involved in competitive sports. Thus, disordered eating prevention programs for the general adolescent population, leisure exercisers and competitive athletes and their coaches should include education about the importance of body appreciation and functionality.

There are some important strengths and limitations of this study that are worth mentioning. Among the strengths of the study is the solid sample of adolescents of both genders representing the cities and the rural regions of Lithuania. Furthermore, this study contributes to growing research on positive body image and adds to the knowledge about its associations with the sport involvement and disordered eating in a highly understudied sample of adolescents. Third, it is the first study providing important knowledge on an adolescent sample of Eastern Europe. The studies of positive body image are very important to countries with the rapid westernisation since a significant portion of adolescents report enormous pressures to attain sociocultural beauty ideals while the efforts to promote positive body image in general schools and sports clubs are at the infant stage [3]. Finally, the results of this study might be important for clinical practice.

Beyond its strengths, this study has some important limitations. The cross-sectional design of this study does not lead to understanding the direction of the associations between the study variables. Therefore, longitudinal studies are recommended to understand causal associations between positive body image and sports participation and the role of body appreciation. The associations between body appreciation and disordered eating in sports involved-adolescents might be mediated by the quality of motivation, with external motivation providing less favourable eating behaviour outcomes [71]. Therefore, future studies should address this issue. The self-reported status of sports participation might also be considered as a limitation, and future studies might benefit from including athletes whom formally participate in professional sports. Finally, we did not assess the level of participation in competitive sports. Elite sports-involved adolescents might significantly differ in body image and dysfunctional eating characteristics; therefore, future studies should address this issue.

## 5. Conclusions

The results of this study support the knowledge of the protective role of positive body image preventing dysfunctional eating in adolescent girls of various sports practice and in competitive sports involved boys. Disordered eating prevention and clinical treatment programs might benefit from including education about body appreciation and functionality. Future studies exploring the underlying mechanisms between the positive body image, sports participation and disordered eating are recommended.

## Figures and Tables

**Table 1 ijerph-17-04044-t001:** Descriptive statistics of the adolescent sample in sports groups (*n* = 1412).

Characteristics		Participation in Sports			χ^2^	*p*
		None (*n* = 346)	Leisure (*n* = 764)	Competitive (*n* = 302)		
Gender (%)	Boys	16.5	54.4	29.1	51.8	<0.001
	Girls	29.9	53.9	16.2		
Body weight (%)	Underweight	16.8	11.7	6.6	25.5	<0.001
	Normal weight	66.7	76.3	83.1		
	Overweight/obesity	16.5	12.0	10.3		
Physical activity score (mean ± SD)		43.7 ± 33.5	69.0 ± 41.3	92.2 ± 47.6	113.3 ^#^	<0.001
Perceived physical fitness (mean ± SD)		2.6 ± 1.0	3.1 ± 0.9	3.8 ± 1.0	131.3 ^#^	<0.001
Exercise, hours/week (mean ±SD)		-	4.6 ± 3.1	7.5 ± 4.3	78.918 *	<0.001
Duration, years (mean ± SD)		-	3.4 ± 3.2	5.1 ± 3.2	−7.9 ^!^	<0.001

*p* = significance level, SD = standard deviation, ^#^ = One-Way ANOVA, F statistics, * = Mann Whitney U test, ^!^ = Independent-Samples *t* test.

**Table 2 ijerph-17-04044-t002:** The comparison of body image, disordered eating behaviours and lifestyle-related factors in sports groups in a sample of boys (*n* = 570).

Characteristics	Non *n* = 94		Leisure Sports *n* = 310		Competitive Sports *n* = 166		Sports Group			Gender		
	M	95% CI	M	95% CI	M	95% CI	F	η^2^	*p*	F	η^2^	*p*
Body appreciation	3.1	2.9–3.3	3.5	3.4–3.6	3.5	3.4–3.7	10.4	0.015	<0.001	9.8	0.007	0.002
Self-esteem	27.3	26.0–28.5	29.5	28.8–30.2	28.6	27.6–29.5	8.7	0.012	<0.001	0.02	0.000	0.892
Body dissatisfaction	1.3	1.2–1.5	1.1	1.0–1.2	1.1	1.0–1.2	4.1	0.006	0.017	111.0	0.07	<0.001
Sociocultural attitudes towards appearance	1.9	1.8–2.1	1.9	1.9–2.0	2.0	1.9–2.1	6.1	0.009	0.002	33.0	0.023	<0.001
SOQ—body functionality	31.0	30.0–32.0	30.7	30.1–31.2	31.3	30.5–32.1	1.2	0.002	0.287	3.2	0.002	0.075
Disordered eating	0.9	0.7–1.1	0.8	0.6–0.9	0.8	0.6–0.9	0.7	0.001	0.503	217.6	0.134	<0.001
Drive for muscularity	2.8	2.6–3.1	3.3	3.2–3.4	3.5	3.4–3.7	173.2	0.110	<0.001	15.1	0.021	<0.001
Perceived physical fitness	2.7	2.5–2.8	3.2	3.1–3.3	3.9	3.7–4.0	115.0	0.141	<0.001	14.1	0.01	<0.001

**Table 3 ijerph-17-04044-t003:** The comparison of body image, disordered eating behaviours and lifestyle-related factors in sports groups in a sample of girls (*n* = 842).

Characteristics	Non *n* = 252		Leisure Sports *n* = 454		Competitive Sports *n* = 136		Sports Group			Gender		
	M	95% CI	M	95% CI	M	95% CI	F	η^2^	*p*	F	η^2^	*p*
Body appreciation	3.0	2.9–3.2	3.3	3.2–3.4	3.2	3.0–3.4	10.4	0.015	<0.001	9.8	0.007	0.002
Self-esteem	27.3	26.6–28.1	28.6	28.0–29.1	29.3	28.3–30.4	8.7	0.012	<0.001	0.02	0.000	0.892
Body dissatisfaction	1.7	1.6–1.8	1.6	1.6–1.7	1.7	1.6–1.7	4.1	0.006	0.017	111.0	0.07	<0.001
Sociocultural attitudes towards appearance	2.1	2.0–2.2	2.2	2.2–2.3	2.4	2.3–2.5	6.1	0.009	0.002	33.0	0.023	<0.001
SOQ—body functionality	29.9	29.2–30.5	30.6	30.1–31.1	30.8	30.0–31.7	1.2	0.002	0.287	3.2	0.002	0.075
Disordered eating	1.6	1.5–1.8	1.8	1.7–1.9	1.9	1.8–2.1	0.7	0.001	0.503	217.6	0.134	<0.001
Drive for muscularity	2.1	1.9–2.2	2.1	2.0–2.2	2.5	2.3–2.7	173.2	0.110	<0.001	15.1	0.021	<0.001
Perceived physical fitness	2.6	2.4–2.7	2.9	2.9–3.0	3.6	3.5–3.8	115.0	0.141	<0.001	14.1	0.01	<0.001

Note: models are controlled for body mass index, M = mean, CI = confidence interval, *p* = significance level, F = Fisher’s statistics, η^2^ = estimate of effect size, SOQ = self-objectification.

**Table 4 ijerph-17-04044-t004:** Comparison of adolescents’ body image concerns, disordered eating and lifestyle-related factors (mean; 95% CI) across weight-sensitive and less-weight-sensitive sports groups.

Characteristics	Weight-Sensitive Sports		Mixed Sports (for Leisure Exercises)		Less Weight-Sensitive Sports		Sports Group			Gender		
	M	95% CI	M	95% CI	M	95% CI	F	η^2^	*p*	F	η^2^	*p*
Body appreciation												
Leisure sports							0.2	0.001	0.827	5.4	0.007	0.02
Boys	3.6	3.4–3.8	3.4	3.2–3.7	3.5	3.3–3.7						
Girls	3.3	3.2–3.5	3.4	3.2–3.5	3.3	3.1–3.5						
Competitive sports							0.2	0.001	0.638	5.5	0.018	0.019
Boys	3.4	3.1–3.7			3.6	3.3–3.8						
Girls	3.3	3.0–3.5			3.0	2.7–3.3						
Self-esteem												
Leisure sports							1.5	0.004	0.232	4.2	0.006	0.041
Boys	30.2	29.1–31.2	29.1	27.7–30.4	29.1	28.0–30.3						
Girls	28.8	28.0–29.6	28.7	27.7–29.8	28.0	26.8–29.2						
Competitive sports							0.3	0.001	0.580	1.1	0.004	0.288
Boys	28.2	26.8–29.6			28.8	27.5–30.1						
Girls	29.2	27.9–30.5			29.7	27.7–31.2						
Sociocultural attitudes towards appearance												
Leisure sports							1.6	0.004	0.207	22.7	0.029	<0.001
Boys	2.0	1.8–2.1	1.9	1.8–2.1	1.9	1.8–2.1						
Girls	2.3	2.2–2.4	2.3	2.2–2.4	2.1	1.9–2.2						
Competitive sports							0.2	0.001	0.624	15.2	0.049	<0.001
Boys	2.0	1.9–2.2			2.1	1.9–2.2						
Girls	2.4	2.3–2.6			2.3	2.1–2.5						
Body dissatisfaction												
Leisure sports							0.9	0.002	0.413	80.4	0.096	<0.001
Boys	0.9	0.8–1.1	1.0	0.8–1.2	1.2	1.1–1.4						
Girls	1.6	1.5–1.8	1.7	1.5–1.8	1.6	1.4–1.7						
Competitive sports							4.7	0.016	0.03	42.4	0.125	<0.001
Boys	1.1	0.9–1.3			1.2	1.0–1.3						
Girls	1.6	1.4–1.7			1.9	1.7–2.1						
SOQ—body functionality												
Leisure sports							1.6	0.004	0.206	0.2	0.000	0.639
Boys	30.8	29.9–31.7	30.3	29.2–31.4	30.7	29.7–31.7						
Girls	30.2	29.6–30.9	30.4	29.5–31.3	31.7	30.7–32.7						
Competitive sports							0.4	0.001	0.512	0.7	0.002	0.416
Boys	31.7	30.6–32.9			30.9	29.9–31.9						
Girls	30.8	29.7–31.9			30.8	29.4–32.3						
Disordered eating												
Leisure sports							0.7	0.002	0.487	145.3	0.161	<0.001
Boys	0.8	0.6–1.0	0.8	0.6–1.1	0.7	0.5–0.9						
Girls	1.8	1.7–2.0	1.8	1.6–2.0	1.7	1.5–1.9						
Competitive sports							0.001	0.000	0.975	88.4	0.229	<0.001
Boys	0.9	0.6–1.1			0.8	0.5–1.0						
Girls	1.9	1.7–2.2			2.1	1.8–2.4						
Drive for muscularity												
Leisure sports							1.1	0.003	0.337	157.1	0.172	<0.001
Boys	3.4	3.2–3.6	3.3	3.0–3.5	3.2	2.9–3.4						
Girls	2.1	2.0–2.3	2.1	1.9–2.4	2.0	1.8–2.3						
Competitive sports							0.003	0.000	0.955	45.8	0.134	<0.001
Boys	3.6	3.3–3.9			3.5	3.2–3.8						
Girls	2.5	2.2–2.7			2.5	2.2–2.9						
Perceived physical fitness												
Leisure sports							6.3	0.017	0.002	21.3	0.027	<0.001
Boys	3.4	3.3–3.6	3.2	3.0–3.4	3.1	2.9–3.2						
Girls	3.0	2.9–3.1	2.9	2.8–3.1	2.8	2.7–3.0						
Competitive sports							8.0	0.026	0.005	4.9	0.016	0.028
Boys	4.0	3.8–4.2			3.7	3.5–3.9						
Girls	3.8	3.6–4.0			3.4	3.1–3.7						

Note: models are controlled for body mass index, M = mean, CI = confidence interval, *p* = significance level, F = Fisher’s statistics, η^2^ = estimate of effect size, SOQ = self-objectification.

**Table 5 ijerph-17-04044-t005:** Factors associated with higher risk of disordered eating in sports groups in a sample of boys (*n* = 570).

Characteristics	Non, *n* = 94			Leisure Sports, *n* = 310			Competitive Sports, *n* = 166		
	B	β	*p*	B	β	*p*	B	β	*p*
Body mass index	0.042	0.181	0.049	0.068	0.220	<0.001	0.048	0.141	0.036
Body appreciation	−0.075	−0.090	0.358	−0.071	−0.090	0.136	−0.193	−0.290	<0.001
SOQ—body functionality	0.011	0.050	0.563	−0.005	−0.028	0.557	−0.026	−0.150	0.031
Self-esteem	0.015	0.090	0.299	−0.010	−0.068	0.246	0.008	0.058	0.413
Body dissatisfaction	0.411	0.311	0.003	0.237	0.189	0.001	0.135	0.117	0.119
Sociocultural attitudes towards appearance	0.502	0.370	<0.001	0.385	0.308	<0.001	0.548	0.448	<0.001
Drive for muscularity	0.094	0.126	0.155	0.012	0.018	0.725	−0.080	−0.127	0.076
Model summary	R = 0.70; R^2^ = 0.48			R = 0.57; R^2^ = 0.33			R = 0.56; R^2^ = 0.32		
F, *p*	11.5; <0.001			20.8; <0.001			10.5; <0.001		

**Table 6 ijerph-17-04044-t006:** Factors associated with higher risk of disordered eating in sports groups in a sample of girls (*n* = 842).

Characteristics	Non, *n* = 252			Leisure Sports, *n* = 454			Competitive Sports, *n* = 136		
	B	β	*p*	B	β	*p*	B	β	*p*
Body mass index	0.021	0.049	0.294	0.039	0.094	0.002	0.066	0.129	0.039
Body appreciation	−0.139	−0.126	0.028	−0.202	−0.165	<0.001	−0.229	−0.194	0.038
SOQ—body functionality	−0.034	−0.148	0.001	−0.019	−0.073	0.013	−0.018	−0.074	0.242
Self-esteem	−0.021	−0.107	0.029	−0.006	−0.028	0.440	−0.009	−0.041	0.573
Body dissatisfaction	0.365	0.271	<0.001	0.511	0.364	<0.001	0.378	0.265	0.005
Sociocultural attitudes towards appearance	0.548	0.350	<0.001	0.573	0.335	<0.001	0.613	0.376	<0.001
Drive for muscularity	0.042	0.038	0.353	−0.032	−0.028	0.332	−0.066	−0.061	0.321
Model summary	R = 0.79; R^2^ = 0.62			R = 0.81; R^2^ = 0.65			R = 0.76; R^2^ = 0.58		
F, *p*	58.0; <0.001			120.1; <0.001			25.1; <0.001		

B = unstandardised regression coefficient, β = standardised regression coefficient, *p* = significance level, SOQ = self-objectification.

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
