# Peer review of "Examining Body Appreciation and Disordered Eating In Adolescents of Different Sports Practice: Cross-Sectional Study"

_ijerph, 2020, doi:10.3390/ijerph17114044_

Round 1

Reviewer 1 Report

This is an important topic, and the manuscript provides valuable additions to the literature. I am especially pleased to see that body acceptance is studied separately from body dissatisfaction. Further citations, details in methods, clarifications to the results, and English language edits are need:

  • Line 21 - should be multiple linear regression analyses were performed
  • line 24 greater than what/whom?
  • Line 45 - I believe this study should be included: Rohde P, Stice E, Marti CN. Development and Predictive Effects of Eating Disorder Risk Factors during Adolescence: Implications for Prevention Efforts NIH Public Access. Int J Eat Disord. 2015;48(2):187-198. doi:10.1002/eat.22270
  • Line 45 - Huge seems like an odd word to use here
  • Line 72 - I would recommend not starting two paragraphs in a row with "However"
  • Here is an article that address the topic of body image among different types of sports:
  • Line 76 - Scoping seems to be an odd word to use here.
  • Line 77 - I would recommend rephrasing to something like, "..higher levels of positive body image" vs "...more positive body image..."
  • Line 82 - is "underreported" the intended word here? under-researched?
  • Line 99 - developmental theory of embodiment was capitalized on the previous page. Keep consistent. 
  • Line 102- consider replacing "functioning" with "impact".
  • Line 120 - This sentence needs a citation.
  • Line - 139 - Did each gym that was invited participate?
  • Line 145 - I am confused? Was recruitment through schools or through gyms?
  • Line 158 - Where was IRB approval obtained from?
  • Line 169 - RSES was not included in parentheses previously. 
  • Line 186 - I am confused...the score ranged from 25 to 25?
  • Were all survey tools validated in Lithuanian?
  • Line 198 - Capitalize the title of the survey tool
  • Line 200 - SATAQ should be included in parentheses in the earlier sentence. 
  • Line 230 - who not whom
  • Line 264-265 - Please clarify that you are referring to time spent weekly, and duration (in years)
  •  Line 280 – The results section is a little hard to follow.
  • Line 285 – This sentence is unclear to me. What does “increased direction” mean?
  • Lin3 298 – Can this paragraph be in a table?
  •  Line 318 – I would prefer that tables stand alone. The values for each gender are not provide in the table; therefore, without the description in the text, it would be unclear which gender group had higher scores. In addition, the font is different from the rest of the text.
  • Line 390 – “Demonstrated” seem to be too strong of a word
  • Line 424 – “Enormously” is an odd word to use here.
  • Line 426 – “To sum it all up” has already been used recently
  • Line 440  - Consider choosing a word other than “enormous”

Author Response

Dear Reviewer,

Thank you very much for your comments. We revised our paper in accordance with your remarks. Corrections are highlighted in the text. Comments and answers table is attached.

Reviewer 2 Report

Title: Examining Body Appreciation and Disordered Eating In Adolescents of Different Sports Practice

The aim of the study was to examine the associations between body appreciation, body functionality and disordered eating in an 1412 adolescent sample of different levels and types of sports practice.

This study is very interesting and I have only minor comments.

Line 130: „(…)controlling for body mass index (BMI].” There is a square bracket at the end – is it a mistake?

Line 144 and 159: subsections should be numbered: 2.1. Procedure and 2.2. Measures

Author Response

Dear Reviewer,

Thank you very much for your comments. We revised our paper in accordance with your remarks. Corrections are highlighted in the text. The comments-answers table is attached.

Reviewer 3 Report

We would like to thank the authors for the consideration of the International Journal of Environmental Research and Public Health for the publication of their manuscript entitled "Examining Body Appreciation and Disordered Eating In Adolescents of Different Sports Practice".

The topic is important for the development of prevention programs for overweight/obesity and eating disorders in adolescents who exercise, either for leisure or competition, that enhance or reinforce a positive body image in adolescents.

Despite the importance of the topic covered and the aspects addressed above, some aspects of the manuscript require revision by the authors for clarity and reproducibility.

Title:

Authors must define the type of study performed in the title as well as in the abstract and methods sections.

Abstract:

In relation to the key words, we suggest that you change "adolescents" to "adolescent". The rest of the keywords do not correspond to MeSH terms. We recommend that the authors replace them (if possible) with appropriate MeSH terms according to the topic of the study.

Introduction:

The introduction should be rewritten, the objective of which is to establish the pillars of the research to be carried out and although this requires a review of the literature, it is not in this section that the results of this review should be discussed, since this should be reserved for the discussion section in which the studies by Romppanen, Biggs and García-Sierra should also be addressed and discussed (in the discussion section).

The objectives should be rewritten according to the change suggested in the abstract/ summary.

Since this is a descriptive study, it does not make much sense to formulate a hypothesis.

Methods:

The methods are not described in sufficient detail and information to replicate the study.

The authors should specify the method of randomization to select the gyms from which the sample was taken.

In addition, they should report on the recruitment process (when, how and by whom) and provide a flowchart to clarify "There were 1,492 students who participated in the study. However, 56 of the students refused to participate by themselves. Furthermore, 24 questionnaires were deleted as they were not filled correctly. For the final analysis, 1,412 questionnaires were confirmed. Perhaps, they meant that 1492 were offered to participate in the study, 56 refused to participate for a total of 1436. However, 24 were excluded from the analysis because they were incomplete. This fragment needs to be reviewed and modified, providing the reasons (if known) why the 56 adolescents refused to participate in the study.

In relation to the measures, what was the question for assessing physical fitness based on? This question has no bibliographical reference. Please provide more information on this. In addition, we suggest that the authors establish a relationship between the variables of the study and the use of each of the tests, due to the high number of measures used.

This section of methods could benefit from the creation of sub-sections for greater clarity and understanding of the text. We recommend that the authors create a statistical analysis sub-section in which they provide the information presented in point 3. Although the data appear to be reliable and statistically sound, we suggest that the authors provide information about who was responsible for handling and performing the analysis and interpreting the data and how this procedure was carried out.

We also suggest that the authors provide another sub-section on ethical and legal aspects, where they expand on the information about the laws by which they were governed and the methods used to ensure confidentiality and data protection. Finally, make a declaration of conflict of interest.

Finally, it is important that the authors mention in the manuscript the reporting guidelines they have followed according to their type of study, in order to improve the quality and transparency of their research. If you have any doubts, please consult the instructions for authors of the journal or https://www.equator-network.org/

Results:

The results must be rewritten even though they are consistent and presented in a complete way to interpret the results.  In the interpretation of the results corresponding to tables 1, 2 and 3, we recommend the authors to support these contributions with the most relevant numerical results obtained. Please bear in mind not to repeat all the information summarized in the tables. However, we suggest the authors to make an additional table to present the results of the gender analysis and the participation in the effects of sport interaction in weight-sensitive versus less weight-sensitive leisure time body image concerns, leaving only the most relevant values in the text.

We recommend the authors to organize the results in sub-sections according to the objective and hypotheses planted for a better understanding of them.

Discussion:

The discussion should be rewritten according to what has been commented on in the Introduction section, in which the results of this research should be contrasted with the results of other research that the authors have already highlighted in their article (see commentary on the introduction above).

Conclusion:

The conclusions seem to be supported by the results and provide a response to the objective and the hypotheses initially raised, providing both statistically significant and non-significant results obtained.

References:

The reference style is appropriate according to the standards for journal authors. Most of the bibliographic references correspond to the last years.

Tables and appendices:

The tables are clear and provide relevant information, without being repeated, according to the content of the manuscript.

The title of appendix A1 should be reviewed, the "n" expressed does not coincide with the values provided in the table. Please clarify this issue.

Author Response

Dear Reviewer,

Thank you very much for your comments and your time reviewing our paper. We corrected our paper in accordance with your remarks. Corrections are highlighted in the text. The comments-answers table is attached.

Sincelery,

The corresponding author

Reviewer 4 Report

Manuscript contains new and significant information adequate to justify publication. The prevention of overweight and obesity and eating disorders  are issues of great importance and very topical. This study aimed to examine the associations between body appreciation, functionality and disordered eating in adolescent sample of different levels and types of sports practice. The research has a large number of participants which is important. Paper demonstrates an adequate understanding of the relevant literature in the field and cites an appropriate range of literature sources. This is well designed manuscript. Methods employed appropriate. Results presented clearly and analysed appropriately. Conclusions adequately tie together the other elements of the paper. Tables and supplementary data are appropriate.

The only complaint is that the introduction is too long, should be more focused with short and concrete sentences.

Author Response

Dear Reviewer,

Thank you very much for your comments and your time reviewing our paper.  The introduction was shortened and revised. Corrections are highlighted in the text.

Round 2

Reviewer 3 Report

Dear authors

We thank you for the new presentation of your modified manuscript ijerph-793236 entitled "Examination of body appreciation and disordered eating in adolescents of different sports practices: cross-sectional study".

Changes made after the reviewers' comments have improved the clarity and reproducibility of their manuscript, as well as their understanding by readers. However, after another specific and careful reading of your contribution, you should note that:

  • Keywords have been changed to free terms, not mesh terms. We suggest that you correct the specific terms "adolescence" and "lifestyle" or explain why there is no mesh term that can replace them, I can think of "adolescent" or "lifestyle", for example.
  • The introduction has been revised and improved. In their responses, they do not provide a justification for why they have considered it relevant to formulate hypotheses, despite being a cross-sectional observational study and in which no association or causality is expected to be evaluated. Please make this reference.
  • In the methods section:
    • They have not clarified the method of randomization. Do it and justify how the centers and participants were randomized, or highlight why it was not done.
    • It is recommended for future research that they collect the reasons why the participants refuse to be part of the study. Such information is important in drawing up a flow chart.
    • Finally, they insist that the authors make reference in this section that the "information guidelines for the observational study in epidemiology" have been followed to carry out the study and write the manuscript.

For my part, the above comments suggest the making of indispensable minor changes before the publication of the manuscript.

Author Response

Dear Reviewer,

We thank you for the specific remarks helping to improve the quality of the manuscript. We revised the manuscript and provide answers to the comments (see in the attachment).
